# *Rhodospirillum rubrum* L-Asparaginase Conjugates with Polyamines of Improved Biocatalytic Properties as a New Promising Drug for the Treatment of Leukemia

Natalia V. Dobryakova [1,2], Dmitry D. Zhdanov [2], Nikolay N. Sokolov [2], Svetlana S. Aleksandrova [2], Marina V. Pokrovskaya [2] and Elena V. Kudryashova [1,*]

1 Chemical Faculty, Lomonosov Moscow State University, Leninskie Gory St. 1, 119991 Moscow, Russia
2 Laboratory of Medical Biotechnology, Institute of Biomedical Chemistry, Pogodinskaya St. 10/8, 119121 Moscow, Russia
* Correspondence: helenakoudriachova@yandex.ru or helena_koudriachova@hotmail.com

**Abstract:** L-asparaginase *Rhodospirillum rubrum* (RrA) is an enzyme (amidohydrolases; EC 3.5.1.1) that catalyzes the L-asparagine hydrolysis reaction to form L-aspartic acid. Due to the shortcomings of existing L-asparaginases from *Esherichia coli* (EcA) and *Erwinia chrysanthemi* (ErA), RrA may turn out to be a new promising drug for the treatment of leukemia. RrA has a low homology with EcA and ErA, which makes the enzyme potentially less immunogenic. RrA has pronounced antitumor activity on a number of leukemia cells. However, there is a need to improve the biocatalytic properties of the enzyme. So, in this study, the RrA conjugates with polyamines with different molecular architectures were developed to regulate the catalytic properties of the enzyme. Linear polyethyleneimine (PEI), branched polyethyleneimine, modified with polyethylene glycol (PEI-PEG), and spermine (Spm) were used to obtain conjugates with RrA. It was discovered by gel permeation chromatography that Spm allows the most active tetrameric form of RrA to be obtained and stabilized. Molecular docking was used to study the binding of spermine to RrA subunits. The activity of the RrA conjugates with Spm and PEI-PEG was 23–30% higher than the native enzyme. The pH optimum of the conjugates shifted from 9.0 to 8.5. The conjugates had higher stability: Spm and PEI-PEG reduced the inactivation constant ($k_{in}$) more than two-fold upon incubation at 53 °C. The conjugate RrA-PEI-PEG reduced the accessibility of trypsin to the protein surface and reduced $k_{in}$ by eight times. The modification of RrA with polyamines made it possible to obtain enzyme preparations with improved biocatalytic properties. These conjugates represent interest for further study as potential therapeutic agents.

**Keywords:** L-asparaginase; *Rhodospirillum rubrum*; activity; spermine; polyethyleneimine; conjugates

## 1. Introduction

L-asparaginase preparations from *Esherichia coli* (EcA) and its PEGylated form, as well as *Erwinia chrysanthemi* (ErA), are currently used in leukemia chemotherapy [1,2]. The principal mechanism of action of the enzymes is associated with their ability to hydrolyze L-asparagine, which results in a decrease in its concentration in the bloodstream. Tumor cells are more sensitive to a lack of L-asparagine, which plays an important role in cell life by participating in the synthesis of proteins and nucleic acids. Contrary to normal cells, cancer cells are often unable to synthesize L-asparagine on their own. As a result, cancer cells undergo apoptosis when this amino acid is deficient. The main drawback of the use of these L-asparaginases in chemotherapy is the hypersensitivity that occurs in patients taking EcA or ErA. To prevent allergic reactions, new types of L-asparaginases derived from other microorganisms can be applied.

One of the promising enzymes for use in the chemotherapy of leukemia is *Rhodospirillum rubrum* L-asparaginase. It has a low homology with EcA and ErA, as well as a shorter amino acid sequence (172 a.a., 18 kDa), which makes the enzyme potentially less immunogenic. The

RrA has pronounced antitumor activity on a number of leukemia cell lines in vitro as well as in vivo [3]. A shifted pH optimum and a high Michaelis constant are the disadvantages of RrA.

One of the methods to improve the physicochemical properties of L-asparaginase is the covalent modification of the enzyme [4]. Modification of EcA with polyethylene glycol (PEG) made it possible to develop a drug with reduced immunogenicity and an increased lifetime in the bloodstream [5,6]. In our earlier works, it was found that chitosan-based copolymers allow us to increase the activity, thermostability, trypsin resistance, and cytotoxicity of *Erwinia carotovora* L-asparaginase (EwA) [7,8]. The mentioned polycations also had a positive effect on RrA. In our previous article, we found that a non-covalent complex formation with polycations such as polyethyleneimine (PEI), PEGylated polyethyleneimine (PEI-PEG), and PEGylated chitosan (chit-PEG) have a positive effect on RrA activity [9]. In particular, 25 kDa of PEI, as part of a polyelectrolyte complex with RrA, increased the rate of L-asparagine hydrolysis for this enzyme by up to 30%. Covalent conjugates of RrA with chit-PEI and chit-PEG also increased the activity by 17 and 32%, respectively. On the other hand, PEI and its copolymers are actively used for the development of anti-cancer drug delivery systems, as well as in gene therapy [10]. Therefore, also focusing on the positive results obtained for RrA, PEI-based polycations are of particular interest for studying further effects on the properties of L-asparaginase. Shorter polyamines, such as spermine and spermidine, also have a cationic nature and have an effect on various biochemical functions of cells [11]. Polyamines are also bifunctional cross-linking agents. They can be used to form covalent cross-links between the subunits of L-asparaginases and influence the activity and stability of the enzyme. Thus, it is of interest in this work to investigate how the cross-linking of spermine affects the properties of RrA.

The aim of the study was to find out how the properties of RrA are affected by conjugation with polyamines based on polyamines: branched PEI-PEG copolymers with an MW of 30 kDa, linear PEI with an MW of 2 kDa, and Spm. Since strong polyelectrolytes can have a negative effect on the enzyme, it is also intended to compare the effect of PEI on RrA compared to its PEGylated form.

## 2. Materials and Methods

### 2.1. Production of the L-Asparaginase Preparation Rhodospirillum rubrum

The RrA enzymes were previously obtained in the laboratory of medical biotechnology at the Institute of Biomedical Chemistry [3]. The RrA gene was isolated from the *Rhodospirillum rubrum* strain (collection of the Microbiology Department, Lomonosov Moscow State University) using a pET23a vector (Novagen) and the following primers: GCCCCTTCC-CTTGCCACAGG, GGACACCCAAGCTTCCCTTTTCCG, and CACAGGATCCTCAAG-GCAAATGGCCG. The active producer was cultivated in Erlenmeyer flasks (the 1 L size) in 200 mL of an LB medium with ampicillin (100 μg/mL) using a GFL 3033 shaking incubator (Germany) at 37 °C. The cell culture density was determined using an Aquarius 7000 spectrophotometer at 600 nm and expressed in optical units (OD600). The inducer (lactose, IPTG) was added to the medium at OD600 of 0.9–1.9 up to the final concentration of 0.2% and 0.001 M, respectively. Afterward, the induction biomass was produced for 17–20 h. At the end of the incubation, the cells were sedimented by centrifugation (15 min, $2500\times g$). The biomass was resuspended in the buffer A (10 mM $NaH_2PO_4$, 1 mM glycine, 1 mM EDTA, pH 7.5) and sonicated in a UZDN2T disintegrator (Russia) for 10 min (1 min of sonication with intervals for 1 min). The cell extract obtained by centrifugation of the sonicated suspension (60 min, $35,000\times g$) was applied onto a QSepharose column ($2.0 \times 30.0$ cm) equilibrated with the buffer A. Fractions containing RrA were diluted 10 times with the buffer A, pooled, and applied onto a DEAE-Toyopearl 650 m column ($1.5 \times 20.0$ cm). In both cases, the protein was eluted using a linear gradient of NaCl concentration (0.0–1.0 M) at the elution rates of 78 and 30 mL/h, respectively. At the final stage, the enzyme solution was desalinated and concentrated using an Amicon cell containing a Millipore filter (NMWL 18,000). All purification procedures were performed at 4 °C.

## 2.2. Synthesis and Purification of L-Asparaginase Conjugates with Polyamines

The following reagents were used for the synthesis: linear polyethyleneimine 2 kDa (PEI, Sigma-Aldrich, St. Louis, MI, USA), branched PEGylated polyethylenimine 30 kDa (PEI-PEG, Sigma-Aldrich, USA), and spermine (Spm, Sigma-Aldrich, USA). The reaction involved the formation of amide bonds between amino groups of polyamines and carboxyl groups of the enzyme in the presence of a Woodward K reagent (Sigma-Aldrich, USA). The RrA preparation was dissolved in 100 mM of a phosphate buffer, pH 6.0, to a final enzyme concentration of 5 mg/mL. Polyamines were also dissolved in the phosphate buffer to molar ratios enzyme:polyamine = 1:10 for PEI, 1:5 for PEI-PEG, and 1:20 for spermine. The polyamine solutions were adjusted to pH 6 if necessary. Then, one of the polyamine solutions was added to the enzyme in the phosphate buffer. To the final mixture with an enzyme concentration of 1 mg/mL, 10 μL of 5 mg/mL Woodward's reagent solution was added. The final mixture was incubated at room temperature with stirring for 2–3 h. The resulting conjugate solutions were purified on Amicon filters (Merck-Millipore, Kenilworth, NJ, USA) with a particle throughput of 3–50 kDa, and the phosphate buffer was replaced with PBS pH 7.5. The purity of the preparation and quaternary composition of the enzyme was controlled by HPLC gel filtration in a Knauer chromatography system (Knauer, Berlin, Germany) on BioFox 17 SEC in a 15 cm × 1 cm$^2$ column. The eluent was 150 mM NaCl (Sigma-Aldrich, USA), pH 7.5; the elution rate was 0.5 mL/min, 25 °C. The resulting conjugates were lyophilized or frozen and stored at 20 °C.

## 2.3. Determination of Zeta Potentials of L-Asparaginase Preparations

Determination of zeta potentials of L-asparaginase preparations was performyd using a "Zetasizer Nano ZS" instrument (Malvern Panalytical, Malvern, UK) in Milli-Q purified water-based solutions containing 1 mg/mL protein. The measurements were performed in a polypropylene cuvette at room temperature. To obtain each zeta potential value, 10 to 15 scans were performed, and the data were averaged.

## 2.4. Registration of CD Spectra

CD spectra of L-asparaginase solutions, as well as purified conjugate solutions, were recorded using a J-815 CD spectrometer (Jasco, Tokyo, Japan) equipped with a thermostatically controlled cell. Measurements were performed in the wavelength range of 200–260 nm at 37 °C in a quartz cuvette (l = 1 mm). Spectra were obtained by 3-fold scanning in steps of 1 nm. An amount of 300 μL of enzyme and conjugate samples in 10 mM PBS pH 7.5 was added to the cuvette. The final protein concentration in the system was 0.5–1 mg/mL.

## 2.5. Registration and Analysis of Infrared Spectra

A Tensor 27 IR Fourier spectrometer (Bruker, Ettlingen, Germany) with an MCT detector was used to obtain the FTIR spectra of RrA and its conjugates. The measurements were carried out in a BioATR II thermostated cell (Bruker, Ettlingen, Germany) using a single reflection ZnSe element at 22 °C and continuous purging of the system with dry air using a compressor, JUN-AIR (Gast Inc., Benton Harbor, MI, USA). An aliquot (40 μL) of the corresponding enzyme solution (0.5–1.0 mg/mL in a 10 mM sodium phosphate buffer) was applied to the internal reflection element, and the spectrum was recorded three times in the range from 3000 to 950 cm$^{-1}$ with a resolution of 1 cm$^{-1}$; we performed 70-fold scanning and averaging. The background was registered in the same way and was automatically subtracted by the program. The resulting spectra were smoothed by the Savitzky–Golay method to a spectral resolution of 2 cm$^{-1}$ [12]. The spectra were analyzed using Opus 7.0 software (Bruker, Ettlingen, Germany). The degree of L-asparaginase modification was determined by the ratio of n(polyamine)/n(enzyme) components. For this purpose, calibration curves were plotted at selected wavelengths for PEI-PEG, PEI (1460 cm$^{-1}$), and Spm (2810 cm$^{-1}$). The wavelength selection criteria were peak reproducibility and linearity of the obtained calibration curves.

### 2.6. Determination of Catalytic Parameters of L-Asparaginase Preparations

The catalytic activity of native L-asparaginase and its covalent conjugates and polyelectrolyte complexes was measured by circular dichroism (CD) using a J-815 CD spectrometer (Jasco, Tokyo, Japan) according to a previously described technique [7]. This method relies on the difference in ellipticity for optically active amino acids, including L-asparagine and L-aspartic acid. This difference allowed us to observe the change in ellipticity over time of substrate hydrolysis by L-asparaginase [13]. L-asparagine (BioChemica, Billingham, UK) at a concentration of 20 mM was mixed with L-asparaginase or its conjugate at concentrations of 0.03–0.035 mg/mL in 10 mM PBS pH 7.5. The reaction was performed in a quartz cuvette with a volume of 300 μL and an optical length of 1 mm in a thermostatically controlled cell at 37 °C.

### 2.7. Determination of pH Dependence of L-Asparaginase Activity

To determine the pH dependence of the native enzyme and its conjugates, we prepared a series of 20 mM L-asparagine solutions in a 5 mM citrate–phosphate–borate buffer. L-asparagine solutions were diluted with a NaOH solution to a pH of 5.5–11. A total amount of 300 μL of the substrate solution in the cuvette and 10 μL of the enzyme with a concentration of 1 mg/mL was added, and the activity was recorded using a CD spectrometer according to the procedure described above.

### 2.8. Thermal Inactivation of L-Asparaginase Preparations

The stability of RrA and its conjugates to thermal inactivation was studied by CD spectroscopy. In a typical experiment, a solution of the enzyme sample (1 mg/mL) in 10 mM PBS (pH 7.5, 100 mM NaCl) was incubated at 53 °C. Every 5–10 min of incubation, aliquots of the native enzyme or PEC solutions were taken and cooled for 4–5 min to room temperature. Then, the catalytic activity of the samples was measured at 37 °C.

### 2.9. Trypsinolysis Resistance of L-Asparaginase Preparations

In a typical experiment, a solution of the native enzyme or conjugate in 10 mM PBS containing 1 mg/mL L-asparaginase and 0.0025 mg/mL trypsin was placed in a thermostat and incubated at 37 °C for 0–60 min. Every 5–10 min, 10 μL of the samples was taken and immediately mixed with a 20 mM L-asparagine solution for further determination of activity by CD spectroscopy.

### 2.10. Autodocking of L-Asparaginase with Spermine

Autodocking was performed with UCSF Chimera version 1.16 (University of California, San Francisco, CA, USA) using the Autodock Vina tool version 1.1.2. [14–16]. The structure of L-asparaginase *Rhodospirillum rubrum* (Q2RMX1) was imported from the website https://www.uniprot.org (accessed on 9 December 2022). SDF structures of L-asparagine and spermine were taken from the website https://www.rcsb.org (Protein Data Bank, accessed on 16 December 2022). Preliminary preparation of the enzyme structure included solvent removal and the addition of hydrogen and charges. Docking was performed first with the dimer form (subunits A and C) of L-asparaginase, then with the tetramer form. The box size for the ligand to search was chosen manually. For clarity, we placed L-Asn separately in the active center. The results were visualized and analyzed using a BIOVIA Discovery Studio Visualizer (https://discover.3ds.com/discovery-studio-visualizer-download, accessed on 11 January 2023). The distance between the amino acid residues, Asp and Glu, between the subunits was calculated in the program PyMol 2.5.2.

## 3. Results

### 3.1. Characteristics of Obtained L-Asparaginase Conjugates

In this study, we obtained three conjugates of L-asparaginase RrA with polyamines PEI, PEI-PEG, and spermine using Woodward's reagent. The reactions' schemes are shown in Figure 1. Table 1 illustrates the characteristics of the polyamines used.

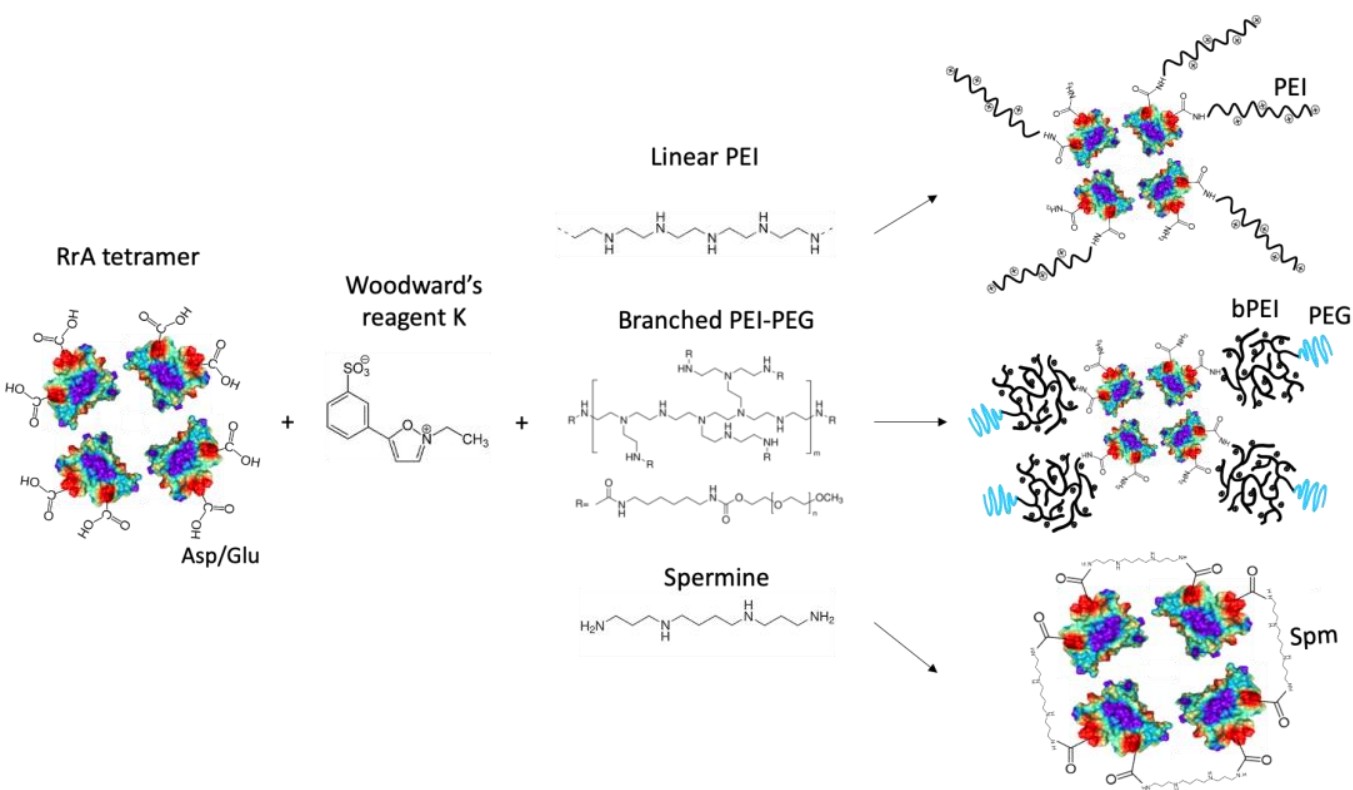

**Figure 1.** The reaction scheme of RrA with Woodward's reagent, polyamines, and resulting conjugates with polyamines.

**Table 1.** The characteristics of the polyamines used in this work.

| Polymer | MM, Da | Origin | Structure | Charge |
|---|---|---|---|---|
| Polyethyleneimine | 2000 | Synthetic | Linear | Polycation |
| PEI-PEG | 30,000 | Synthetic | Branched | Polycation |
| Spermine | 202 | Natural | Linear | Polycation |

The obtained conjugates were characterized by gel filtration, CD, and IR spectrometry.
Figure 2 shows the chromatograms of the RrA solutions and the RrA-PEI-PEG and RrA-Spm conjugates. The protein fractions were detected at 280 nm. The retention times ($t_R$) corresponding to the major peaks on the chromatogram of the native RrA and its modified forms are shown in Table 2. The native enzyme has a main peak at approximately 24 min and a small peak at 18 min. These two peaks correspond to dimeric and tetrameric forms (36 and 72 MM), which follows a comparison with the control sample, BSA, which has an MW of about 66.5 kDa. Thus, RrA is represented in tetrameric and dimeric forms in the solution. As we can see, the dimeric form is predominant (24 min). The conjugation of RrA with PEG-PEI shifts the equilibrium between the two forms toward the tetrameric form. It is confirmed from the ratio of peaks on the conjugate chromatogram, where the ratio of the peaks at $t_R$ 17.6 and 23.9 min is approximately 1:1. The free PEI-PEG copolymer has two small peaks at 24.7 and 26.7 min. The conjugation of RrA with spermine leads to a shift in the equilibrium toward the tetrameric form. A single peak corresponding to the tetrameric form is observed in the RrA-Spm chromatogram.

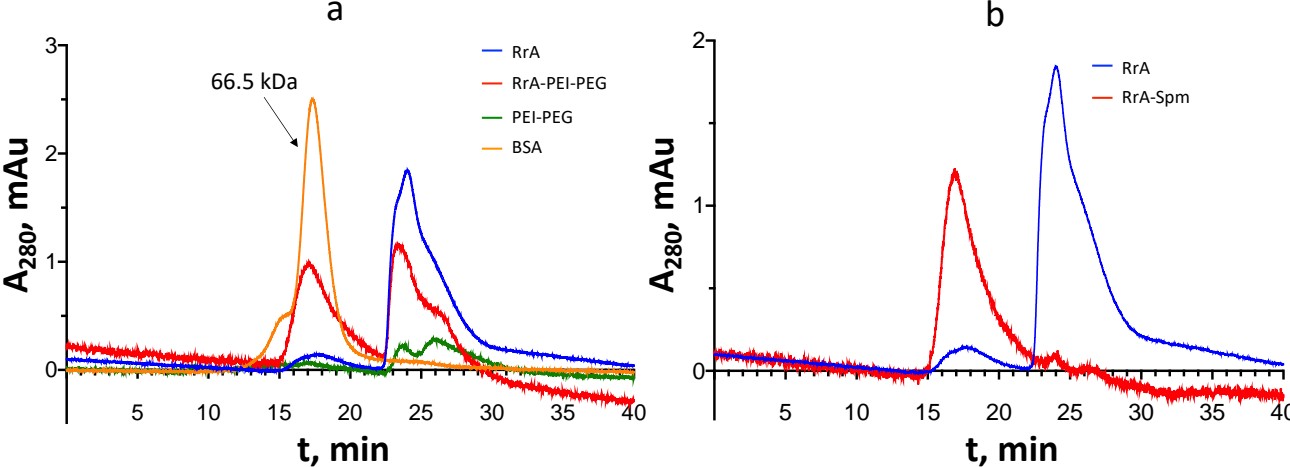

**Figure 2.** Chromatograms obtained by gel filtration: (**a**) solutions of RrA and conjugate, RrA-PEI-PEG; (**b**) native RrA and conjugate, RrA-Spm. Protein concentration of the samples is 1 mg/mL; C(PEI-PEG) = 5 mg/mL.

**Table 2.** Retention times for the main fractions of native RrA solutions and their conjugates.

| RrA Preparation | $t_R$, Min | | Control Samples | $t_R$, Min |
| --- | --- | --- | --- | --- |
| | Tetramer | Dimer | | |
| RrA | 18.3 | 24.3 | BSA | 17.5 |
| RrA-PEI-PEG | 17.6 | 23.9 | PEI-PEG | 24.7; 26.7 |
| RrA-Spm | 17.3 | - | | |

Figure 3 shows the IR spectra of the purified solutions of the RrA-PEI-PEG, RrA-Spm, and RrA-PEI conjugates. The spectra of the unmodified enzyme, as well as the covalent conjugate of RrA with the PEI-PEG copolymer, show characteristic peaks of amide I at 1500–1600 cm$^{-1}$ and amide II at 1600–1700 cm$^{-1}$, corresponding to the absorption of the amide bonds of the proteins. Additionally, the conjugate has absorption bands at 1460 cm$^{-1}$ (N-H vibrations of the secondary amino groups and CH$_2$ bonds) and 1070 cm$^{-1}$ (C-O-C bond vibrations), characteristic of the PEI-PEG copolymer (Figure 3a). The RrA-Spm conjugate also has bands in the amide I and amide II regions (Figure 3b). It has peaks in the 1035–1230 cm$^{-1}$ region, corresponding to the C-N bond vibrations in the polyamine molecule. In addition, some contribution to the intensity of the amide I peak can be made by the band at 1620 cm$^{-1}$, corresponding to the N-H bond vibrations of the amines. RrA-PEI (Figure 3c) has an intense, broad band at 1050–1150 cm$^{-1}$. A band at 1450–1640 cm$^{-1}$ (N-H, CH$_2$) can also be distinguished in both the polymer and the conjugate.

The obtained conjugates were analyzed using CD spectrometry for the content of the secondary structures in comparison with the native enzyme. Figure 4 shows the CD spectra of the native enzyme and the conjugates with PEI, PEI-PEG, and Spm. The shape of the spectrum characteristic of the native enzyme is preserved for the conjugates, as well. The content of the secondary structures changes slightly: in the conjugates, the number of alpha-helices decreases, and the percentage of disordered structures increases (Table 3). The percentage of antiparallel beta structures, which are responsible for aggregation, increases [17].

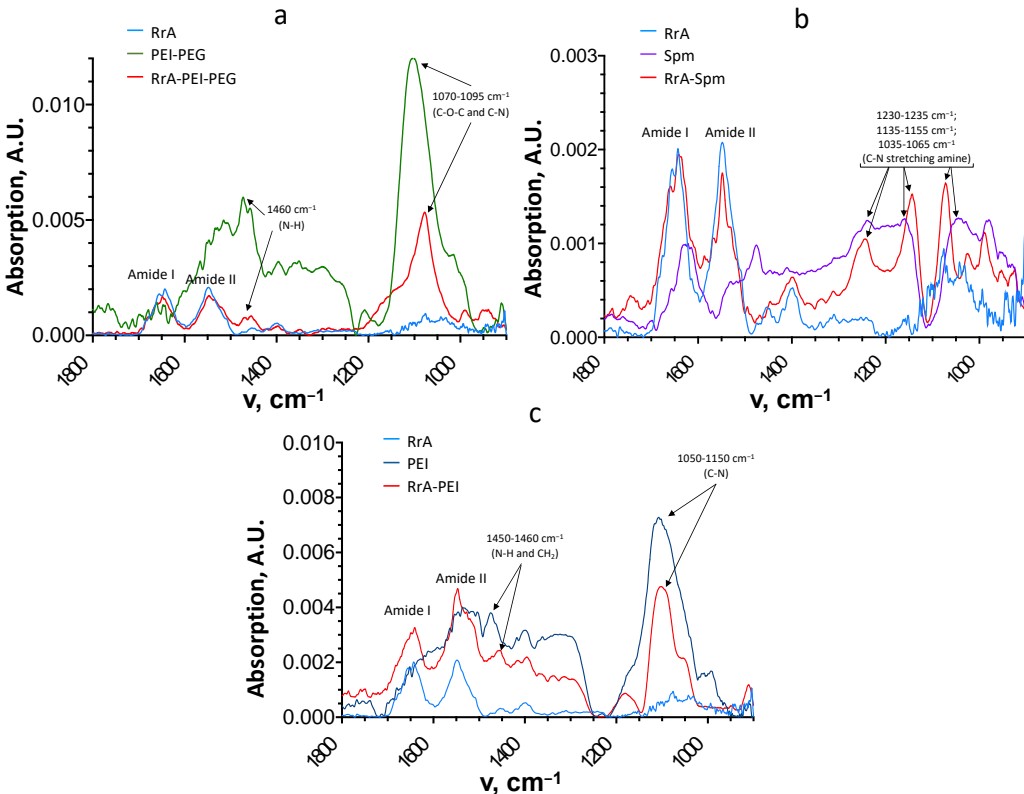

**Figure 3.** IR spectra of the native enzyme compared with conjugates: (**a**) RrA-PEI-PEG; (**b**) RrA-Spm; (**c**) RrA-PEI. The concentration of RrA was 1 mg/mL, PEI-PEG was 1 mg/mL, Spm was 0.5 mg/mL, and PEI was 0.1 mg/mL. Conditions: PBS 10 mM, 22 °C.

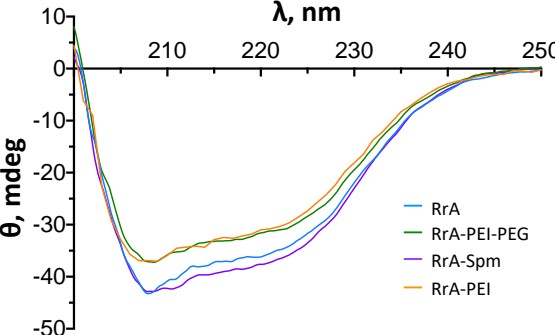

**Figure 4.** CD spectra of the native enzyme and RrA-PEI-PEG, RrA-PEI, and RrA-Spm conjugates. C(RrA) = 1 mg/mL, 0,01 M PBS, 37 °C, pH 7.5.

**Table 3.** Percentage content of native enzyme secondary structure elements and RrA-PEI-PEG, RrA-PEI, and RrA-Spm conjugates, calculated in the CDNN program.

| Secondary Structures | RrA | RrA-PEI | RrA-PEI-PEG | RrA-Spm |
|---|---|---|---|---|
| Helix | $35.4 \pm 1.1$ | $31.0 \pm 1.0$ | $31.4 \pm 0.9$ | $32.1 \pm 1.0$ |
| Antiparallel | $7.6 \pm 0.7$ | $9.0 \pm 0.7$ | $8.8 \pm 0.6$ | $8.6 \pm 0.7$ |
| Parallel | $8.7 \pm 0.3$ | $9.3 \pm 0.3$ | $9.3 \pm 0.2$ | $9.1 \pm 0.2$ |
| Beta-turn | $17.1 \pm 0.3$ | $17.3 \pm 0.4$ | $17.2 \pm 0.3$ | $17.1 \pm 0.3$ |
| Random coil | $31.5 \pm 1.1$ | $34.0 \pm 1.0$ | $33.9 \pm 1.2$ | $33.3 \pm 0.9$ |

To characterize the obtained conjugates and find correlations between the surface charge and enzyme activity, we measured the particle surface charge and specific activity (Table 4). The polyamine modification increases the enzyme charge from −9.0 to +1.4 mV

(for RrA-PEI-PEG) compared with the native protein. All of the conjugates had a higher specific activity than the native enzyme. The conjugate with the PEI-PEG copolymer showed the greatest increase in activity. The conjugate with a linear PEI showed an insignificant increase in activity by about 5%. The RrA-Spm conjugates showed an activity increase of 23%. Polyamines, therefore, have a positive effect on the specific activity of L-asparaginase and a minimal effect on the secondary structure of the enzyme.

**Table 4.** Zeta potential and specific activity values of native RrA and its conjugates. The hydrolysis activity of 20 mM L-Asn was measured in 10 mM PBS pH 7.5 at 37 °C.

| RrA Preparation | Degree of Modification * | Zeta Potential, mV | Specific Activity, IU/mg |
|---|---|---|---|
| RrA | NA | $-9.0 \pm 1.3$ | $47 \pm 4$ |
| RrA-PEI-PEG | 2 | $1.4 \pm 0.3$ | $61 \pm 4$ |
| RrA-PEI | 6 | $-4.0 \pm 1.8$ | $49 \pm 2$ |
| RrA-Spm | 18 | $-8.1 \pm 0.9$ | $58 \pm 3$ |

* Degree of modification = n(polyamine)/n(enzyme).

Next, to determine how conjugation affects the properties of the enzyme, several parameters of the conjugates were studied in comparison with the native enzyme. These parameters were the pH optimum of the enzyme, thermostability, trypsinolysis resistance, and cytotoxic activity.

### 3.2. Docking of L-Asparaginase with Spermine

To explain the effect of the spermine conjugation on the enzyme equilibrium shifted from the dimer toward the tetrameric form of RrA, we modeled the interaction of the dimer and tetramer of RrA with spermine using the UCSF Chimera and Autodock vina programs. We also calculated the interatomic distances between the closest carboxyl groups, Asp and Glu (Table 5). The table illustrates the possible pairs of amino acids that can form electrostatic interactions with spermine [18,19]. When the carboxyl groups on the amino acids are modified with Woodward's reagent, spermine, a bifunctional agent, can form covalent bonds between these amino acid pairs. Spermine has a molecular length of about 16.5 Å. This means that it is most likely to form bonds between amino acid pairs whose distance between them does not exceed this value.

**Table 5.** Interatomic distances between Asp and Glu residues for the RrA subunits. Pairs of amino acids that can potentially form two-point cross-links are marked with "+". The pairs that turn out to be closest to spermine as a result of docking are highlighted in bold.

| Pair of Amino Acids | Distance, Å | Spermine (16.1 Å) |
|---|---|---|
| Asp123–Glu166 | 15.8 | + |
| Glu166–Glu166 | 10.3 | + |
| Glu166–Asp169 | 19.7 | +/− |
| Asp123–Asp123 | 11.2 | + |
| Glu25–Asp79 | 21.1 | +/− |
| Glu25–Glu24 | 17.5 | + |
| Glu24–Glu24 | 20.9 | +/− |
| Asp152–Asp170 | 12.4 | + |
| Glu166–Asp170 | 12.0 | + |
| Asp152–Asp152 | 16.0 | + |
| Asp169–Asp88 | 18.4 | +/− |
| Asp169–Asp123 | 20.3 | +/− |
| Glu25–Asp20 | 24.4 | |
| Glu24–Asp20 | 16.5 | + |
| Glu24–Glu37 | 25.1 | |

There are almost no bonds between the active center and spermine when different sites of the protein are selected, including the whole molecule, according to the results of docking with the dimer and tetramer (Figure 5). Spermine should not interfere with the hydrolysis of L-asparagine, which is consistent with the obtained activity results. When docking with the RrA dimer, spermine is located close to at least five Asp and Glu pairs from Table 5. However, only three are suitable for cross-linking formation: between Asp170 and Glu166, between Asp152 and Asp152, and between Asp169 and Asp88. In the case of the tetramer, the number of potential contacts increases. So, the simulation of the spermine interaction with the tetramer realizes six of fifteen possible contacts. Indeed, additional contacts between Glu24 and Asp20, Glu166 and Glu166, and Glu166 and Asp169 emerged in the case of the tetramer. In the case of a dimer, one RrA subunit interacts with only one side of the other subunit. In the case of a tetramer, an additional interaction surface is formed between the subunits. In this case, additional amino acids of each monomer are involved in the interaction. The model does not show the real distance between the subunits, so other amino acid pairs cannot be excluded. We can assume that spermine, as a highly active cross-linking agent, will tend to form more bonds in the tetramer than in the dimer. The experimental results suggest that spermine stabilizes the tetrameric structure.

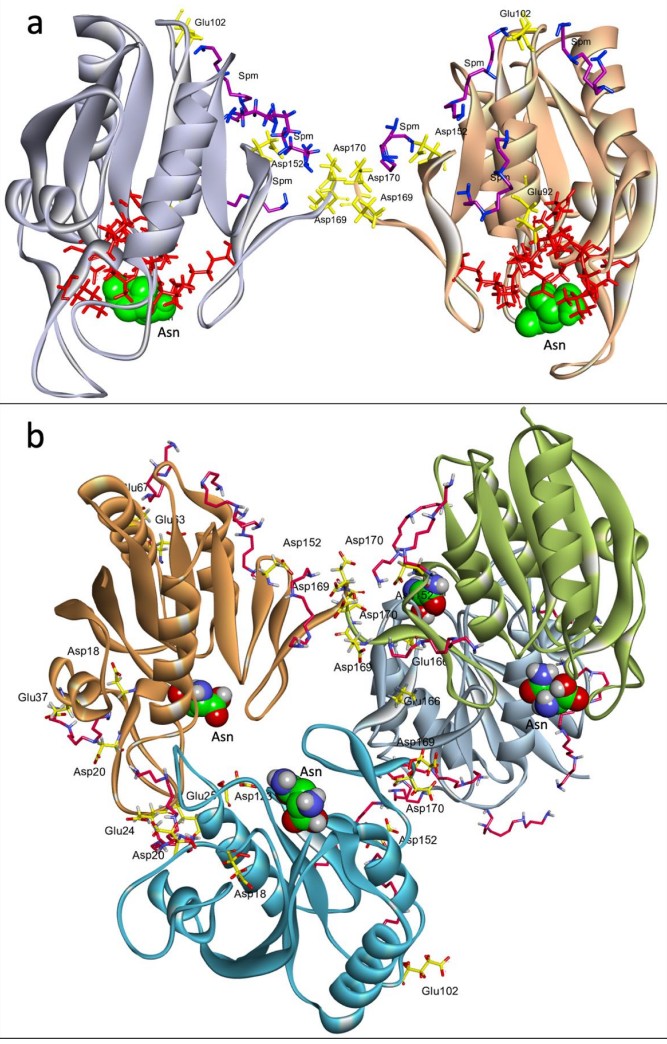

**Figure 5.** Results of docking: (**a**) dimer RrA with spermine and L-Asparagine; (**b**) tetramer RrA with spermine and L-Asparagine. Structure of L-asparaginase *Rhodospirillum rubrum* imported from website https://www.uniprot.org (Q2RMX1, accessed on 9 December 2022). Asp and Glu residues are highlighted in yellow, spermine molecules are highlighted in lilac, substrate L-asparagine is highlighted in green (atomic balls), and residues included in the enzyme active center are highlighted in red.

### 3.3. Dependence of the Activity of RrA and Its Conjugates on pH

To determine the effect of polyamines on the pH optimum of the enzyme, we measured the activity of native RrA and its conjugates at a pH of 5.5–11.0. The obtained dependencies are shown in Figure 6.

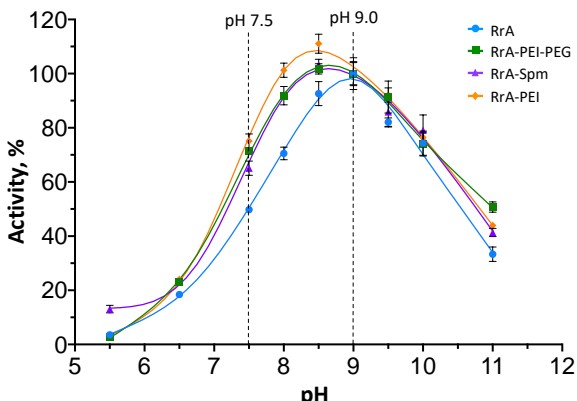

**Figure 6.** pH dependence of activity for native RrA and its conjugates: 5 mM citrate phosphate–borate buffer, 37 °C.

Both the native enzyme and the conjugates have a typical bell-shaped dependence with a maximum pH of 8.5–9.0. The conjugates RrA-PEI, RrA-PEI-PEG, and RrA-Spm have a 0.5-unit shift to a more acidic region compared to the native enzyme.

At a pH of 7.5, the activity of the conjugates increases by 13–22% compared to the native enzyme. As in the PBS, the RrA-PEI-PEG conjugate was most active at a pH of 7.5 in the citrate phosphate–borate buffer (CPBB). The native enzyme had an activity of 60 IU/mg at a pH of 7.5 in the CPBB. RrA-PEI-PEG, RrA-Spm, and RrA-PEI had an activity of 74, 71, and 68 IU/mg, respectively. At the same time, the activity of RrA and its modified forms at a pH of 8.5–10 was lower than that of native RrA.

### 3.4. Thermostability of RrA Enzyme Preparations

The effect of conjugation of the enzyme with the polyamines on the thermostability of RrA during incubation at 53 °C was studied. At close to this temperature, the enzyme undergoes a phase transition, resulting in changes in the protein structure. The native enzyme loses more than 50% of its original activity in 60 min. The obtained curves of thermal inactivation and semilogarithmic dependence of the residual activity on time are shown in Figure 7.

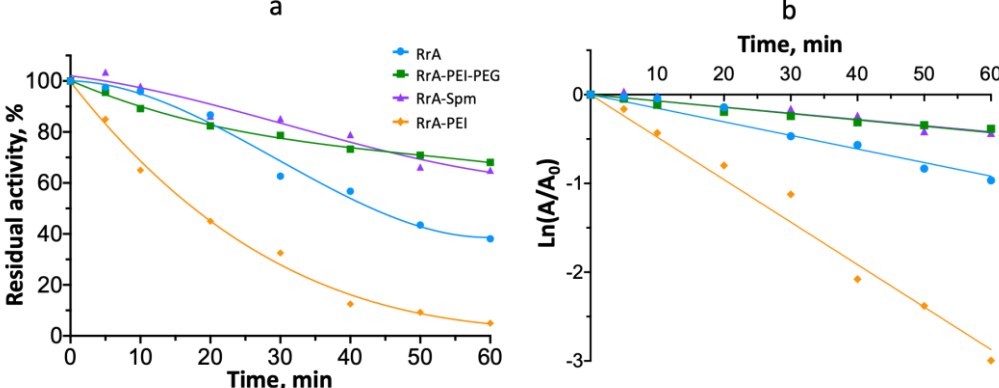

**Figure 7.** (**a**) Thermal inactivation curves of the residual activity of native RrA and its conjugates after incubation at 53 °C for 60 min; (**b**) semilogarithmic time dependencies. The concentration of the enzyme for all samples was 1 mg/mL. A is the specific activity of the enzyme at time t, and $A_0$ is at t = 0.

The polyamines studied have different effects on the stability of RrA. Table 6 shows the values of the inactivation constants and the percentage of residual activity after the incubation of the samples for 60 min. The residual activity of RrA-PEI-PEG and RrA-Spm was approximately two-fold higher than that of the native enzyme and was 68 and 65%, respectively, whereas that of the native enzyme was 38%. Thus, PEI-PEG and Spm promoted the increased stability of native RrA at elevated temperatures. At the same time, the RrA-PEI activity drops to 5% of the original activity, which is eight times lower than that of the native enzyme. The resulting curves are well described by the first-order inactivation equation. The values of the first-order inactivation constants ($k_{in}$) were obtained from the slope angle of the lines in semi-logarithmic coordinates (Figure 7b). The inactivation constants for the RrA conjugates with PEI-PEG and Spm are more than two-fold lower than for the native enzyme. For the RrA-PEI conjugate, $k_{in}$ was 0.05 min$^{-1}$, which is three-fold higher than that of the native. Linear, unmodified PEI has a destabilizing effect on the enzyme and does not contribute to maintaining the quaternary structure of the protein.

**Table 6.** Values of the first-order inactivation constants of RrA and its conjugates.

| RrA Preparation | $k_{in}$, min$^{-1}$ | Residual Activity, % |
|---|---|---|
| RrA | $0.017 \pm 0.002$ | 38 |
| RrA-PEI-PEG | $0.006 \pm 0.001$ | 68 |
| RrA-Spm | $0.008 \pm 0.001$ | 65 |
| RrA-PEI | $0.050 \pm 0.004$ | 5 |

### 3.5. Stability of Enzyme Preparations to Trypsinolysis

An important parameter for characterizing the obtained RrA conjugates is their resistance to trypsinolysis. This parameter correlates with immunogenicity because it shows the availability of the enzyme to the plasma proteins and the components of the complement system. The time at which the enzyme retains its catalytic activity in the presence of trypsin is increased when RrA is modified with polyamines. The resulting inactivation curves for the trypsinolysis reaction, as well as the semilogarithmic relationships for determining the inactivation constants, are shown in Figure 8. The values of the constants are given in Table 7.

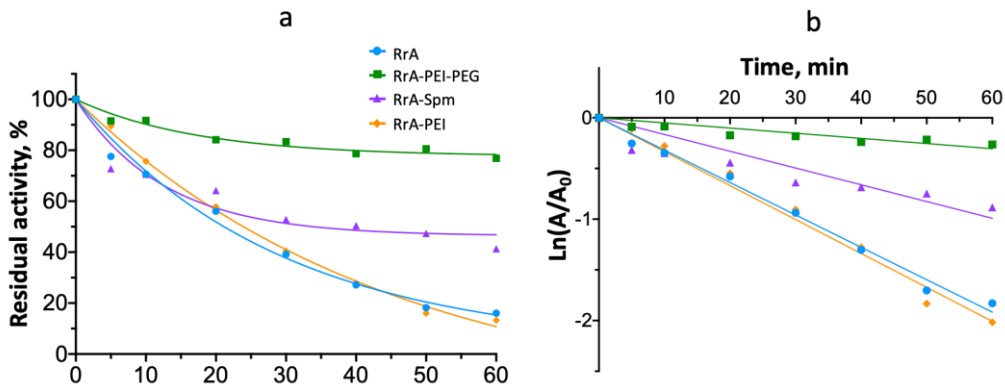

**Figure 8.** Trypsinolysis curves (**a**) and semi-log dependences (**b**) of the residual activity of native RrA and its conjugates on the incubation time at 37 °C. The concentration of L-asparaginase in the samples was 1 mg/mL; the concentration of trypsin was 0.0025 mg/mL; PBS pH 7.5.

**Table 7.** Inactivation constants in the trypsinolysis reaction for native RrA and its conjugates.

| RrA Preparation | $k_{in}$, min$^{-1}$ |
|---|---|
| RrA | $0.031 \pm 0.003$ |
| RrA-PEI-PEG | $0.004 \pm 0.001$ |
| RrA-Spm | $0.012 \pm 0.002$ |
| RrA-PEI | $0.035 \pm 0.002$ |

The data obtained show that the RrA-PEI-PEG conjugate with a branched structure is the most resistant to the action of trypsin compared to the native RrA and other conjugates. Its $k_{in}$ was 0.031 min$^{-1}$. The inactivation constant of RrA-Spm was also lower than that of the native enzyme and was 0.012 min$^{-1}$. PEI had the least effect: the kin value for the RrA-PEI conjugate was 0.035 min$^{-1}$, which is close to the constant for the native enzyme, 0.031 min$^{-1}$. Linear PEI probably attracts trypsin due to polyelectrolyte interactions, so stabilization was not observed.

## 4. Discussion

We proposed the modification of L-asparaginases with polycations as a promising method to improve their biopharmaceutical properties. From the results obtained, we can see that polyamines affect the activity, structure, and stability of the enzyme.

With HPLC, we were able to see that RrA is present in the solution in two different forms, dimeric and monomeric (Figure 2). It was previously known that L-asparaginases of the mesophilic bacteria *Escherichia coli* and *Erwinia carotovora* could exist in dimeric and tetrameric forms [20]. In this case, the active site of L-asparaginase is located at the junction of two subunits, so the monomer has no catalytic activity. Figure 2 shows that the covalent modification of RrA can stabilize the tetrameric form. Spermine probably covalently binds the protein oligomers together and allows for a fully tetrameric form of RrA, which presumably exhibits greater activity than the dimer. Indeed, RrA-Spm has greater activity than native RrA. The PEI-PEG also shifts the equilibrium toward tetrameric, but to a lesser extent, because of the longer, branched chain of PEG-PEI. RrA-PEI-PEG activity also increases, which is probably due to the formation of a tetramer.

In the case of the interaction with the dimer and tetramer of RrA, the most likely and energetically advantageous binding sites of spermine were identified using molecular docking. It is known that oligoamines, including spermine, can stabilize and increase the activity of enzymes when forming noncovalent complexes. For example, it was shown on chymotrypsin that polyelectrolyte complexes, with oligoamines, increase the activity of the enzyme and protect it from aggregation in the water–organic phase [18]. At the same time, the effect of the increased enzyme activity was observed only at high molar concentrations of the oligoamines, exceeding the enzyme concentration by ~100 times. We also did not observe any particular effect of increased activity in the formation of the protein–polyelectrolyte complexes of spermine and spermidine with RrA at concentrations below 0.25 mM (data not shown). In addition, the large number of multipoint interactions with the amino groups can corrupt the native conformation and destabilize the enzyme. Spermine can also increase the activity and thermal stability of lysozyme [21]. The binding of the lysozymes and alpha-globulin with spermine can be related to the interaction of oligoamine with asparagine and glutamine amino acid residues. In another study, the effect of spermine on the properties of myoglobin was evaluated [22], where it was also shown that spermine binds to the protein molecule and increases its stability. Using molecular docking, it was shown that the amino groups of spermine form hydrogen bonds with the carboxyl groups of the amino acid residues, Asp and Glu. In addition, spermine influenced the secondary structure of the protein by decreasing the percentage of alpha-helices and increasing the content of the beta structures.

Electrostatic bonds, however, are highly dependent on the solvent and its ionic strength and may be unstable when the drug is administered intravenously. Therefore, obtaining the most active stable tetrameric form will provide a potential drug for use in therapy.

This increased activity is also influenced by the local pH of the solution where the enzyme is located. The increase in the activity of RrA-PEI-PEG at a pH of 7.5 is related to the polycationic nature of the copolymer. PEI-PEG shifted the pH optimum of the enzyme to a more acidic region by changing the pH near the active center of the enzyme. For chitosan-PEG and other chitosan copolymers, the same results were previously shown. Spermine also increased the RrA activity by approximately 17% at a pH of 7.5–8.0. This is probably due to the stabilization of the tetrameric form of the enzyme, the ratio of which

can change with changes in the pH of the solution. At an alkaline pH, PEI decreases the enzyme activity more than the other polyamines (up to 25%) and slightly increases the activity at a pH of 7.5 (by 13%). As mentioned earlier, this polycation can destabilize the enzyme due to its higher positive charge density.

The loss of enzyme activity at elevated temperatures can be explained by two factors: dissociation of the enzyme to the monomers and denaturation of the protein sequence. When the temperature is increased, dissociation of the enzyme subunits is usually observed, which leads to the separation of the amino acids that make up the active center and the loss of activity. Therefore, it is logical that RrA-Spm loses activity approximately two-fold slower at an elevated temperature than the native enzyme since there is covalent cross-linking of the monomers to each other. As expected, PEI has a destabilizing effect on RrA, accelerating the decrease in activity. PEI is longer than spermine, and this prevents the oligomeric forms of the protein from cross-linking with each other. Moreover, the high concentration of positive charge can negatively affect the interaction of the enzyme subunits, destroying the hydrogen bonds between them. In PEI-PEG, a portion of the primary amino groups is modified by PEG, so conjugation with this copolymer avoids the negative effect of a large amount of positive charge on the quaternary structure of the protein.

The obtained conjugates also showed a different resistance to the trypsin action, which is related to the immunogenicity of the enzyme preparation—the availability of the enzyme to the proteins of the complement system. In the previous work, it was shown that chitosan copolymers, due to their branched structure and shielding of the protein globule, allow us to protect the enzyme from degradation by proteases [9]. Inactivation constants for the conjugates with chit-PEG and chit-PEI with a branched structure decreased four- and nine-fold, respectively. In the current work, we also found that branched PEI-PEG increased the resistance of RrA to trypsinolysis more than the other conjugates. PEI had almost no effect on the resistance of RrA to trypsinolysis, which may also be due to the destabilization (a disordered tertiary structure) of the enzyme. In addition, the charged end chains of PEI may, on the contrary, promote binding to the proteases, reducing the resistance to the trypsin action.

Earlier, it was shown that covalent modification of EwA with chitosan copolymers allows for increasing the enzyme activity, increasing its thermal stability and resistance to trypsinolysis, and cytotoxicity [7,8,23]. The RrA modification with these copolymers also improved the same physicochemical parameters and had almost no effect on the secondary structure of the protein [9,24]. However, it was found that chitosan-PEG increased the RrA activity no more than 1.5-fold. For EwA, such modification turned out to be much more effective: the enzyme activity increased 3–4-fold depending on the degree of PEGylation of the chitosan. The different effects of the chitosan copolymers can be explained by the fact that EwA and RrA differ in their structure and properties. These enzymes have only about 25% peptide sequence homology and differ from each other in molecular weight, particle charge in a neutral solution, hydrolysis activity, and affinity for L-asparagine and L-glutamine [25]. It was also shown in a previous article that the formation of the polyelectrolyte complexes of RrA with PEI increases the activity of the enzyme by about 30% [9]. In the case of EwA, PEI had the opposite effect, reducing the hydrolysis activity of L-asparagine by several times. It is evident that the studied polyelectrolytes have different effects on the properties of L-asparaginases from different sources.

## 5. Conclusions

Polyamines turned out to be promising substances for improving the physicochemical properties of RrA. The PEGylated form of PEI promoted higher activity and a gentler effect on the enzyme compared with unmodified PEI. Spermine promoted the stabilization of the tetrameric form of the enzyme and increased its activity and stability. Thus, the obtained conjugates of RrA with polyamines may be interesting as improved drugs in leukemia therapy.

**Author Contributions:** N.V.D. and E.V.K. contributed to the conception and design of the study. M.V.P. and S.S.A. performed the molecular cloning and purification of the enzymes. N.V.D., D.D.Z. and N.N.S. conducted the experiments. E.V.K. and D.D.Z. contributed to the data analysis and interpretation of the data. N.V.D. and E.V.K. drafted the article. E.V.K. and D.D.Z. critically reviewed and commented on the manuscript. All authors have read and agreed to the published version of the manuscript.

**Funding:** The work was performed in the framework of the Russian Federation's fundamental research program for the long-term period from 2021 to 2030 (№ 122022800499-5).

**Institutional Review Board Statement:** Not applicable.

**Informed Consent Statement:** Not applicable.

**Data Availability Statement:** The data presented in this study are available on request from the corresponding author.

**Acknowledgments:** The work was performed using equipment (FTIR spectrometer Bruker Tensor 27 and Jasco J-815 CD Spectrometer) of the program for the development of Moscow State University.

**Conflicts of Interest:** The authors declare no conflict of interest.

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
