# Peer review of "Rhodospirillum rubrum L-Asparaginase Conjugates with Polyamines of Improved Biocatalytic Properties as a New Promising Drug for the Treatment of Leukemia"

_applsci, doi:10.3390/app13053373_

Round 1

Reviewer 1 Report

Dear authors

The manuscript could be published after correcting spelling and grammatical errors.

Sincerely

Reviewer 2 Report

Reviewer’s Comments:

The manuscript “Development of L-asparaginase Rhodospirillum rubrum preparations with improved properties by formation of polyamine conjugates with different molecular architectures” is a very interesting work. In this work, L-asparaginase Rhodospirillum rubrum (RrA) is an enzyme (amidohydrolases; EC 3.5.1.1) that catalyzes L-asparagine hydrolysis reaction to form L-aspartic acid and ammonia. Due to the shortcomings of existing preparations of L-asparaginases from Esherichia coli (EcA) and Erwinia chrysanthemi (ErA), RrA may turn out to be a new promising drug for the treatment of leukemia. Compared to existing drugs, RrA has a shorter amino acid sequence, high stability, and comparable cytotoxicity against leukemia cells in vitro. However, there is a need to improve the physicochemical parameters of the enzyme. In this study, the activity and stability of RrA as well as its conjugates with polyamines were investigated. Linear polyethyleneimine (PEI), branched polyethyleneimine modified with polyethylene glycol (PEI-PEG), and spermine (Spm) were used to obtain conjugates with RrA. It was discovered by gel permeation. While I believe this topic is of great interest to our readers, I think it needs major revision before it is ready for publication. So, I recommend this manuscript for publication with major revisions.

1. In this manuscript, the authors did not explain the importance of the polyamine conjugates in the introduction part. The authors should explain the importance of polyamine conjugates.

2) Title: The title of the manuscript is not impressive. It should be modified or rewritten it.

3) Correct the following statement “The RrA-PEI-PEG conjugate with the most branched structure reduced trypsin access to the protein and reduced kin 8-fold. The obtained conjugates made it possible to obtain an enzyme preparations with improved physicochemical characteristics. These enzymes will be of interest for further study as potential therapeutic agents”.

4) Keywords: The keywords should be small. So, modify the keywords.

5) Introduction part is not impressive. The references cited are very old. So, Improve it with some latest literature like 10.3390/molecules27196457, 10.3390/biom12010083

6) The authors should explain the following statement with recent references, “The table illustrates the pairs of amino acids that can form electrostatic interactions with spermine..

7) Please justify the following statement “In the case of the tetramer, the number of contacts increases. Contacts are added between Glu24 and Asp20, Glu166 and
Glu166, Glu166 and Asp169”
.

8) The author should provide reason about this statement “In contrast, RrA-PEI-PEG and RrA- Spm stabilize the structure of L-asparaginase”.

9) Comparison of the present results with other similar findings in the literature should be discussed in more detail. This is necessary in order to place this work together with other work in the field and to give more credibility to the present results.

10) Conclusion part is very long. Make it brief and improve by adding the results of your studies.

11) There are many grammatic mistakes. Improve the English grammar of the manuscript.

Reviewer 3 Report

This research article entitled “Development of L-asparaginase Rhodospirillum rubrum preparations with improved properties by formation of polyamine conjugates with different molecular architectures” by Natalia V. Dobryakova and co-workers is an intriguing part of research in the field of leukemia therapy. Polyamines proved to be promising substances for improving RrA's physicochemical properties. Thus, polyamine modification of RrA and investigation of the properties of these conjugates are of great interest. As a result, I recommend this manuscript for publication after minor changes.

1. The Abstract should be more concise, with clearly stated results.

2. Some sentences lost their meaning. See per example lines 39-41.

3. The information given in Table 1. should be presented in Materials and Methods.

4. Check typing errors. See per example line 304.
